# Disrupted Calcium Homeostasis in Duchenne Muscular Dystrophy: A Common Mechanism behind Diverse Consequences

**DOI:** 10.3390/ijms222011040

**Published:** 2021-10-13

**Authors:** Barbara Zabłocka, Dariusz C. Górecki, Krzysztof Zabłocki

**Affiliations:** 1Molecular Biology Unit, Mossakowski Medical Research Institute Polish Academy of Sciences, 02-106 Warsaw, Poland; bzablocka@imdik.pan.pl; 2School of Pharmacy and Biomedical Sciences, University of Portsmouth, St Michael’s Building, White Swan Road, Portsmouth PO1 2DT, UK; 3Military Institute of Hygiene and Epidemiology, 01-163 Warsaw, Poland; 4Laboratory of Cellular Metabolism, Nencki Institute of Experimental Biology Polish Academy of Sciences, 02-093 Warsaw, Poland

**Keywords:** Duchenne muscular dystrophy, calcium signalling, calcium homeostasis, mitochondria, endoplasmic reticulum

## Abstract

Duchenne muscular dystrophy (DMD) leads to disability and death in young men. This disease is caused by mutations in the *DMD* gene encoding diverse isoforms of dystrophin. Loss of full-length dystrophins is both necessary and sufficient for causing degeneration and wasting of striated muscles, neuropsychological impairment, and bone deformities. Among this spectrum of defects, abnormalities of calcium homeostasis are the common dystrophic feature. Given the fundamental role of Ca^2+^ in all cells, this biochemical alteration might be underlying all the DMD abnormalities. However, its mechanism is not completely understood. While abnormally elevated resting cytosolic Ca^2+^ concentration is found in all dystrophic cells, the aberrant mechanisms leading to that outcome have cell-specific components. We probe the diverse aspects of calcium response in various affected tissues. In skeletal muscles, cardiomyocytes, and neurons, dystrophin appears to serve as a scaffold for proteins engaged in calcium homeostasis, while its interactions with actin cytoskeleton influence endoplasmic reticulum organisation and motility. However, in myoblasts, lymphocytes, endotheliocytes, and mesenchymal and myogenic cells, calcium abnormalities cannot be clearly attributed to the loss of interaction between dystrophin and the calcium toolbox proteins. Nevertheless, DMD gene mutations in these cells lead to significant defects and the calcium anomalies are a symptom of the early developmental phase of this pathology. As the impaired calcium homeostasis appears to underpin multiple DMD abnormalities, understanding this alteration may lead to the development of new therapies. In fact, it appears possible to mitigate the impact of the abnormal calcium homeostasis and the dystrophic phenotype in the total absence of dystrophin. This opens new treatment avenues for this incurable disease.

## 1. Introduction

Duchenne muscular dystrophy (DMD) is a debilitating and lethal neuromuscular disorder. Diagnosis is made usually between the ages of two and five. The earliest symptoms include waddling gate, frequent falling, and difficulties with climbing stairs. However, studies of human foetuses [1,2,3] and in various animal models [4,5,6] revealed that pathology starts already in the prenatal development. The first DMD defects are detectable in developing mesenchymal cells, before their differentiation into muscle [7]. Subsequently, muscle transcriptomes from asymptomatic DMD patients reveal typical dystrophic abnormalities [8], and developmental delays are present in 2-month-old DMD babies [9].

Progressing skeletal muscle weakness and wasting eventually lead to the loss of ambulation (around the age of 10–12) and to the respiratory impairment, the latter exacerbated by frequent infections. Patients require ventilation at around the age of 20. Cardiac muscle complications are usually the late symptom [4], suggesting that dystrophic striated muscle wasting may not be directly related to its workload. Eventually, the respiratory and/or cardiac failure develop, leading to the premature death between 20 and 40 years of age [10]. This range reflects the differences in the supportive care available to different patients’ populations rather than the advances in the disease modifying therapies.

Although fatal consequences of DMD result from muscle damage, DMD also affects other organs and tissues. The most important are CNS and bone defects, the first leading to cognitive and behavioural impairments, the latter causing skeletal deformities exacerbating muscle symptoms.

### 1.1. Genetics of Duchenne Muscular Dystrophy

DMD is caused by the out-of-frame mutations within the *DMD* gene. As the gene locus is on the X chromosome, DMD sufferers are mainly boys, with an incidence of 1 per 5000 male births, but 1 in 50 million females are also affected [11]. *DMD* is the largest human gene known, spanning 2.22 Mb in the Xp21 region. It consists of 79 exons, and it is driven by eight promoters. Of the identified DMD mutations, ~70% are exonic deletions, <15% duplications, and 18–20% are small mutations, with 75% being the nonsense/frameshift types, with splice-site mutations constituting ~20% [12]. One-third of all these are de novo mutations, which precludes the standard prenatal diagnosis from preventing the occurrence of the disease [13].

Importantly, there is an allelic variant of the disease, a mild form known as Becker muscular dystrophy (BMD), with an incidence between 1 in 18,000 and 1 in 30,000 live male births. The clinical picture of BMD is similar to DMD, yet symptoms begin later and progress at a much slower rate. The differences in clinical phenotypes, despite mutations affecting the same gene, appear to be caused by the mutation type: The loss-of-frame mutations resulting in the complete absence of dystrophin cause DMD, while mutations preserving the reading frame and thus allowing for the production of partially functional mini-dystrophin proteins result in BMD [14].

Transcription of human dystrophin may be initiated at eights promoters. Three of these drive the expression of the “full-length” dystrophin transcripts, translated into 427 kDa dystrophin proteins. These differ at their N-termini by only 3, 7, and 11 amino acids, but are expressed in a tissue-specific manner. The remaining five promoters are positioned across the gene and control expression of N-terminally truncated dystrophins (Dp412, Dp260, Dp140, Dp116 and Dp71) [15].

Of note, the genotype–phenotype correlation is not unambiguous [12,16]. The *DMD* gene mutation hotspots are located in the regions affecting the full-length isoforms. Indeed, mutations leading to the loss of full-length 427 kDa dystrophin are both necessary and sufficient for causing the muscle pathology [17,18]. The CNS abnormalities seem to be further exacerbated by the loss of the short isoforms Dp71 and Dp140, which are highly expressed in the normal brain [19]. There are few data on the muscle impact of mutations affecting these short isoforms or on the dystrophin-null patients. In this review, we have therefore focussed primarily on the canonical DMD caused by the absence of Dp427. Interestingly, genotype–phenotype correlation studies identified genetic modifiers [20], which can influence the disease progression in patients with the same *DMD* gene mutation e (see below).

### 1.2. Dystrophin Structure and Presumed Functions

The full-length dystrophin in striated muscle is located under the sarcolemma. It contains several domains with specific functions: Its N-terminus interacts with actin cytoskeleton while the C-term contains domains which bind a set of dystrophin-associated proteins assembling into the subsarcolemmal and transmembrane dystrophin-associated protein (DAP) complex. Thus, dystrophin connects the actin cytoskeleton to the DAP complex in the cell membrane and DAPs, in turn, act as receptors for specific extracellular matrix proteins. Moreover, additional binding sites across the large rod domain allow interactions with further scaffolding and signalling proteins (reviewed in [21]). Particularly relevant to this overview are the direct and indirect interactions with proteins involved in calcium homeostasis such as plasma membrane calcium ATPase (PMCA), calcium channels TRPC1 and TRPC4, calmodulin, and calcineurin. These arrangements have been described based mostly on the muscle studies [22], but they exist in other cells too. Importantly, the composition of the DAP complex, and perhaps also its functions, can vary in different tissues and cells and even in different regions of the cell. These differences, where relevant to the topic of this review, will be discussed in detail.

Importantly, the mechanism(s) triggering muscle death have not been fully elucidated, although “the sarcolemmal hypothesis” put forward in 1990s has become the leading one and it weighs heavily on the therapeutic approaches. It stems from the findings on the dystrophin-DAP interactome abnormalities and can explain many clinical findings. This dominant view on the pathogenesis of DMD states that in the absence of dystrophin anchoring the DAP, this complex is not properly attached in the myofiber sarcolemma and that its loss destabilises the cell membrane, which becomes torn during contractions. Moreover, impaired architecture of intracellular scaffolds affects the appropriate localisation and function of various other proteins important for myofiber operations [23].

Yet, several studies, rather unexpectedly, challenged this widely accepted view of dystrophin anchoring DAP in the myofibre sarcolemma being a key to muscle health. It has been shown that ablation of dystrophin in fully differentiated myofibres did not trigger DAP dysregulation or muscle degeneration [24], while ablation of both dystrophin and DAP only triggered a mild decrease in muscle contractile force but not muscle degeneration [25]. Astoundingly, there is even evidence that myofibres can function entirely without dystrophin, just with overexpression of specific genetic modifiers [26]. Furthermore, cell-autonomous defects are noticeable in DMD cells, which, when healthy, express the 14 kb DMD transcript but do not produce detectable levels of full-length dystrophins [27,28,29,30]. These, undoubtedly surprising findings, shed a new light on DMD pathophysiology (see below).

It is also important to remember that dystrophin deficiency leads to a variety of cell-specific symptoms, which are the sum of the loss of functions conveyed by specific isoforms in a cell-specific context. This variability may, at least partially, reflect the unique subcellular localisations of dystrophins in different cell types: in muscle fibres, dystrophin is evenly distributed beneath the sarcolemma, while in neurons it forms puncta limited to the postsynaptic regions [31,32,33,34,35].

Given this complex picture, it is remarkable that mutations affecting a range of dystrophins and occurring in different cells can still produce common abnormalities. In particular, in all dystrophic cells, whether in patients or animal models and whether muscle or non-muscle, one pathomechanism appears to be always present: the DMD-evoked changes in calcium homeostasis.

The characteristic feature of DMD is that tissues and organs of dystrophic individuals have exceedingly high cytosolic calcium ion concentrations, both upon stimulation as well as in resting conditions. This omnipresent feature of DMD pathology has been identified long before the genetic basis has been discovered and observed not only in patients but also in as distinct models of this disease as *mdx* mouse and zebrafish and also in isolated dystrophic cells [36,37,38,39]. Moreover, altered calcium homeostasis is found in dystrophic cells which, when healthy, produce Dp427 but also in those that express transcripts encoding it but do not have detectable levels of dystrophin protein, which loss could explain the abnormalities.

## 2. Objectives of This Review

Understanding the disrupted calcium homeostasis, the consistent alteration amongst the plethora of defects in this complex pathology, is important. It might be the one defect underlying all the other.

The purpose of this review is to (i) discuss the diverse aspects of calcium abnormalities across the affected tissues, as the aberrant mechanisms leading to it have cell-specific components, (ii) compare the obvious but overlooked disconnect between pathological mechanisms in cells where dystrophin can be a scaffold for proteins engaged in calcium homeostasis with cells in which calcium alterations cannot be clearly attributed to the loss of interaction between dystrophin and the calcium toolkit, and (iii) evaluate the existing therapeutic approaches for their impact on correcting the calcium overload in dystrophic cells.

We describe the current knowledge but also highlight the clear inconsistencies in our understanding of this dystrophic anomaly. Recognising these discrepancies is critically important for understanding past failures and developing new, better therapies.

## 3. Mechanisms of Calcium Homeostasis

Calcium ions play a universal and versatile role in all cells and, in fact, it would be difficult to find a completely calcium-independent cellular process [40]. Quiescent, healthy cells, electrically excitable as well as non-excitable ones, maintain the bulk cytosolic calcium ions concentration ([Ca^2+^]c) at the level of 100–120 nM, while extracellular [Ca^2+^] level is approximately 1.0–1.8 mM [41,42,43].

Therefore, calcium fluxes through the plasma membrane and between organelles, as well as calcium storage within specific intracellular structures, must be precisely regulated and adjusted to specific demands of a particular cell and to its physiological state. General mechanisms are common for all metazoan cells, both electrically excitable and non-excitable ones and involve a relatively limited spectrum of molecules known as the calcium toolkit (Figure 1). It contains proteins responsible for activation of calcium signals (receptors, Ca^2+^ channels in the plasma membrane (PM), the endoplasmic reticulum (ER), acidic calcium stores (for review see in [44]) and proteins which transmit calcium signals between organelles participating in calcium responses, such as ER and PM, calcium sensing effectors, which execute cellular response indirectly (i.e., calmodulin, annexins, S-100 proteins) or directly (i.e., enzymes such as calcineurin, PKC, calpains, phospholipase A, MLCK, and many others), and finally proteins, which terminate calcium responses restoring the resting cytosolic concentration.

### Calcium Signalling Toolkit 

Ca^2+^ entry requires activation of a variety of cell-specific calcium channels including voltage-gated, ligand-gated, stretch-activated, receptor-activated, and capacitative (store-operated calcium entry (SOCE)). The latter is preceded by a depletion of the ER stores of calcium and engages Stim proteins in the ER membrane and Orai (and/or TRP-family member) proteins located in PM. Ca^2+^ release from ER occurs through ryanodine receptors (RyRs) or IP3-activated receptors located in the ER membrane. Both channels are also regulated by Ca^2+^. Generation of IP3 follows activation of specific receptors in PM and activation of PLC with participation of a specific G protein. Mitochondria may transiently accumulate Ca^2+^, but only if the local Ca^2+^ concentration is substantially increased (i.e., in a close vicinity of calcium channels in the PM or ER membranes). ER contains Ca^2+^ binding proteins of high capacity and low affinity (calsequestrin and calreticulin). It is the major Ca^2+^ store in animal cells. Ca^2+^ is removed from the cytosol to the extracellular fluid against its concentration gradient (exploiting PMCA and sodium/calcium exchangers (NCX)) and to the ER stores (with sarco/endoplasmic calcium ATPase (SERCA)).

Although general principles of cellular calcium homeostasis and the set of molecular tools which are engaged in Ca^2+^ transport, storage and decoding calcium signals are basically the same in all cells, spatial organisation, interactions, and regulation of activity differ among various cell types. Figure 2 shows, in a very schematic way, specific arrangement of proteins which interact upon excitation of skeletal muscle.

Calcium signals are triggered upon opening of Ca^2+^ channels in the plasma membrane, which allows Ca^2+^ entry, or due to activation of specific receptors in the ER membranes resulting in calcium release from these stores. In both cases, Ca^2+^ flux is driven by the electrochemical potential of this cation. These processes are very fast and do not need any additional energy input. In contrast, a restoration of the resting, post-stimulatory cytosolic Ca^2+^ concentration is an energy-consuming process. The latter relies on Ca^2+^ removal through the plasma membrane and/or its accumulation within the endoplasmic reticulum. In ER, calcium is bound and buffered by specific, low affinity and high capacity Ca^2+^-binding proteins (i.e., calsequestrin and calreticulin) and stored in an easily available form (Figure 1). Ca^2+^ removal from the cytosol to the extracellular space and to ER occurs against its concentration gradient, thus it needs energy delivered by ATP hydrolysis (for calcium pumps) or at the expense of the electrochemical potential of another ion simultaneously transported in the opposite direction (i.e., using sodium/calcium exchangers, NCX). Cytosolic calcium binding proteins (i.e., parvalbumin in skeletal muscle) and mitochondria, which may transiently accumulate Ca^2+^, modulate the intensity of calcium transients by locally buffering excess of Ca^2+^ ions and influencing calcium-dependent processes (such as Ca^2+^ release/uptake from/into ER or the store-operated calcium entry). Moreover, mitochondria are the major source of cellular ATP, which is required to cover energy demands of calcium pumps and ion exchangers.

Highly dynamic calcium homeostasis is maintained due to an orchestrated interplay between proteins composing the calcium toolkit. As their relative amounts, intracellular distribution as well as some regulatory and kinetic properties may differ in various cells, the final profiles of calcium responses (duration, amplitude, and frequency of calcium waves and oscillations) are not the same and undergo precise, cell-specific modulations. On the other hand, under the resting condition, cytosolic Ca^2+^ levels are basically similar in all cells. However, a highly ordered network of mutual interactions involving positive and negative feedback mechanisms responding to the actual local Ca^2+^ concentration within a particular cellular compartment seriously impedes attempts at identification of the primary consequence of changes in a protein, which does not belong to the calcium toolkit. Yet, such a protein not directly engaged in the cellular calcium handling may still seriously influence Ca^2+^ homeostasis. Dystrophin, as it will be explained, seems to be such a protein.

## 4. Abnormal Calcium Homeostasis in DMD

Loss of the *DMD* gene expression appears to disturb the normal cytosolic Ca^2+^ level, which reflect the dynamic equilibrium between Ca^2+^ entry into and its removal from the cytosol. *This physiological balance appears altered in all DMD affected cells.*

Persistently elevated resting cytosolic Ca^2+^ concentrations and abnormally high calcium transients reflect an aberrant intracellular calcium homeostasis and are a direct cause of severe cellular damage and organ dysfunctions. First, they evoke mitochondrial calcium overload, altered cellular energy metabolism and activation of calcium-dependent enzymes, including PLA2 and calpains. These enzymes catalyse hydrolytic degradation of phospholipids and proteins, respectively [46,47]. Eventually, calcium overload may result in an activation of apoptosis and/or other modes of cell death [48,49]. Such pathological scenarios have indeed been found to occur in most of the dystrophic cells. Therefore, DMD defect may be considered as a generalised cellular calcium deregulation, where phenotypic consequences of altered calcium depend on individual, tissue-specific factors.

## 5. Calcium Dysregulation in DMD Skeletal Muscle

As the progressive muscle impairment is the cause of premature death of DMD patients, most studies have been focused on this tissue and relate to differentiated skeletal muscle fibres.

Myofibres have a unique mechanism responsible for large and reproducible increases in the cytosolic calcium concentration, which are crucial for muscle contraction. The electrical or chemical excitation of myofibres depolarises the sarcoplasmic membrane and, in consequence, triggers an interaction of the L-type voltage gated calcium channel/dihydropyridine receptor (LTCC/DHPR) located in the so-called T-tubules formed by sarcoplasmic membranes with the ryanodine receptors (RyR) in the SR membrane (Figure 2). This assembly of membranes is known as a triad [50]. Subsequently, Ca^2+^ is released from SR to the cytoplasm and activates muscle contraction. Skeletal muscle fibres may contract for a long time in a Ca^2+^-free environment due to the recycling of Ca^2+^ released from the SR stores upon cell excitation. It has been found that lack of dystrophin affects LTCC/DHPR–RyR interaction in skeletal muscle upon their stimulation [51,52].

### Dystrophin, the Scaffold

The functional or structural link between dystrophin and specific proteins more or less directly involved in calcium signalling has been suggested quite early, but a precise mechanism of such interactions has remained unknown [30,43,53,54,55,56,57].

The earliest theory explaining calcium overload in dystrophic muscle fibres had implicated destabilisation and rupture of the sarcolemma in the wake of muscle contraction. Indeed, the leaking out of intracellular creatine kinase and uptake of Evans blue dye into damaged fibres and sarcolemmal “tears” (the so-called ‘delta’ lesions) described in early electron microscopy studies of dystrophic muscles [58,59] seem to support such a mechanism. However, while these findings certainly indicate severe damage of the sarcolemma of dystrophic pre-necrotic fibres, they are not evidence of its causation [60]. Therefore, while still recalled, this concept has been supplanted by other hypotheses.

Functions of the DAP complex in muscle are not limited to the mechanical strengthening and stabilisation of the sarcolemma. DAP is a multiprotein molecular scaffold, also for proteins, which belong to the calcium toolkit and are directly involved in the activation, modulation, or termination of calcium signals [61,62]. A significant proportion of almost 200 proteins found to be altered in the sarcolemma of *mdx-4cv* dystrophic mice (lacking the full-length dystrophin) were proteins involved in the regulation of ion homeostasis [63]. The DAP misassembly appears to affect calcium signalling at the sarcoplasmic membrane level [63].

While calcium channel blockers have little impact on the DMD pathology [64], the mechanisms downstream from the channel entry seem to affect the calcium response that directly controls the excitation-contraction coupling. Whereas expression of mini-dystrophin in *mdx* muscle restored Ca^2+^ entry through channels [51], mass spec analysis of dystrophin and DAP in sarcoplasmic membrane compartments failed to confirm a direct interaction of dystrophin with the triad complex [45]. Therefore, alternatively, dystrophin and DAP may modulate the interaction between L-type Ca^2+^ channels and RyR [51,65].

An excessive increase in cytosolic Ca^2+^ concentration in muscle cells during contraction and failure to return to the normal calcium level prevents muscle fibres from being able to relax. This deficit in muscle relaxation is a specific DMD feature across species l [66].

In myotubes differentiated from human induced pluripotent stem cells (hiPSCs) Shoji et al. (2015) showed that the electric stimulation-evoked contraction caused a pronounced calcium ion influx in DMD cells [67]. On the other hand, dystrophin re-expression corrected cytosolic Ca^2+^ concentration [56,57] and prevented its efflux from SR through the inositoltrisphosphate receptor (IP3R) [43]. This observation indicates that, in dystrophic muscle cells, the excessive Ca^2+^ release via IP3R located in SR may cause its abnormal elevation in the cytosol and contribute to the imperfect muscle relaxation. Similarly, oxidative stress-related modifications of the RyR protein in dystrophic muscle result in the enhanced Ca^2+^ leakage from the SR stores, leading to permanently elevated cytosolic calcium levels [68]. In addition, an aberrant architecture and motility of ER due to lost interactions between dystrophin and actin, and thereby between actin cytoskeleton and the ER membranes, may influence the local cytosolic calcium concentration: Ca^2+^ is taken up or released by the neighbouring ER stores. Elevated Ca^2+^ release from ER is probably not the only cause of the prolongation of the post-excitatory phase of the calcium signal. A slowed down reuptake of Ca^2+^ due to decreased SERCA activity, observed in microsomes from dystrophic human muscles, extends this post-stimulatory elevation of cytosolic Ca^2+^ [69,70]. On the other hand, slower Ca^2+^ release from SR and the altered SR organisation lead to dispersed Ca^2+^ puffs instead of those being localised at the triads [71]. Such a dysregulation of Ca^2+^ distribution within cells results in rises of Ca^2+^ in regions that are usually not exposed to such an excessive stimulation.

Abnormally high intracellular Ca^2+^ causes undesirable activation of hydrolytic enzymes [72] and this may result in irreversible damage [73].

## 6. Calcium Abnormalities in DMD Myogenic and Non-Muscle Cells

While the calcium-based regulation of skeletal muscle contraction is a unique phenomenon, muscle cells also use other, less tissue-specific mechanisms. These are shared with other cells, both electrically excitable and non-excitable. Cells essential for muscle regeneration and thus particularly important for the disease are myoblasts. Unlike myofibres, in these electrically non-excitable cells, release of Ca^2+^ from SR does not lead to contraction upon stimulation of LTCC/DHPR but other mechanisms of calcium signalling become activated instead.

### The Loss of the Absent

Until recently, potential DMD abnormalities in myoblasts were not investigated. The reason for this was that healthy myoblasts do not express detectable dystrophin protein. Therefore, no abnormalities have been foreseen. Against this reasonable prediction, in proliferating myoblasts, mutations in the DMD gene cause multiple changes in calcium signalling [28,30]. Next-generation sequencing (NGS) and subsequent biochemical analyses revealed changes in the expression of a number of genes encoding calcium signalling and related proteins (e.g., STIM 1, SERCA1 and 2A, calsequestrin, PLC variants 3 and 4, NCX 1 and 2, PMCA, Gαq11, and IP_3_R) [28]. Moreover, numerous functional abnormalities including elevated SOCE activity [30], altered responses of both metabotropic and ionotropic receptors [28,29], and reduced oxygen consumption, together with stimulated glycolysis [74] distinguish immortalised *mdx* myoblasts from their wild type (w/t) equivalents. As these cells were maintained for generations in culture, changes appear to be cell-autonomous rather than triggered by the dystrophic niche. On the other hand, corresponding experiments in primary myoblasts confirmed key results obtained in immortalised cells [30,75], thus excluding these alterations as being long-term culture effects. However, identification of the primary calcium response component directly affected in dystrophic myoblasts proved difficult so far, mainly because all elements of the “calcium puzzle” are interconnected and mutually dependent. Nevertheless, these findings complement the growing body of experimental data showing that biochemical consequences of *DMD* gene mutations, including the aberrant calcium homeostasis, are detectable at the very early stages in muscle cell differentiation, and long before manifestation of any significant clinical symptoms.

In fact, in chimeras with incorporation of just 10–30% of *mdx* embryonic stem cells (ESC), the heart and skeletal muscle displayed severe DMD phenotypes, with isolated cardiomyocytes showing increased calcium response [76]. Myogenic precursors obtained from DMD hIPSc exhibited the DMD phenotype upon differentiation towards the skeletal muscle lineage [7]. Furthermore, DMD lymphoblasts showed enhanced purinergic sensitivity [27] and dystrophic endotheliocytes are also dysfunctional [77,78,79].

These findings indicate that a phenotype can appear in a range of cells that, when healthy, are not known to express dystrophin proteins. Clearly, this poses an important question regarding the mechanism behind this alteration. All cells which present the phenotype express the full-length dystrophin transcript when healthy, and it is significantly depleted in dystrophic cells. Note that this 14 kb mRNA has a transcription time of ~16 h [80]. Its expression in rapidly dividing myoblasts is therefore unlikely to be “illegitimate”. Translation and a precise spatio-temporal requirement for small amounts of the full-length dystrophin, analogous to its role in satellite cells [81] or at neuronal synapses [34], require further studies. Another possibility is a mechanism not involving a protein product. Considering the increasing number of human pathologies caused by RNA-mediated disease process, one could speculate that the loss of a large 14 kb transcript might trigger abnormalities [reviewed in [82]. Their presence in *mdx* cells, that harbour a point mutation, indicate that the mechanism does not involve large gene rearrangements. In turn, improvements resulting from the expression of mini-dystrophins (see below) do not support toxic effects of its breakdown products. However, the impact of premature dystrophin transcript termination on the misregulation of other genes cannot be excluded [83].

Some recent data favour alterations of the epigenetic regulation as a more likely mechanism. Absence of dystrophin/DAPC in satellite cells leads to aberrant epigenetic control, which impairs functions of the “offspring” myoblasts that appear to be harbouring somatically heritable epigenetic changes [75].

It could be argued that restoration of selected functions with mini-dystrophin expression or via exon skipping contradicts the importance of *DMD* gene expression in undifferentiated muscle cells. However, the mini-dystrophin transgene transfection and exon skipping in vivo are particularly effective in proliferating cells, such as myoblasts. Therefore, effects of these treatments might result from dystrophin re-expression in myoblasts with their subsequent differentiation into dystrophin-positive myotubes.

Taken together, these data show a continuum of DMD pathology that starts in development [7], affecting myogenic stem cells, persists in myoblasts and critically affects the functional development of differentiating myotubes [67]. As a result, dystrophic myofibres are unable to withstand the contraction-induced injury, while satellite cell and myoblast dysfunctions prevent normal regeneration, further exacerbating the pathology.

Importantly, data from other tissues further support the early, developmental nature of this pathology and confirm the key role of calcium alterations in the pathogenesis of DMD.

## 7. Calcium Abnormalities in DMD Cardiac Muscle

Although the regulatory mechanism of cardiomyocyte contraction differs from that in skeletal muscles, it also relies on changes in the cytosolic calcium concentration. The common feature of skeletal muscle and cardiac myocytes is high dystrophin abundance. However, its interaction with respective calcium toolkits is significantly different. In contrast to skeletal muscles, cardiomyocytes do not form triads and the heart muscle cannot contract in the absence of extracellular Ca^2+^. Ca^2+^ release from ER is based on the calcium-induced calcium release (CICR) mechanism. It involves activation of RyR by Ca^2+^ entering cells through voltage-gated channels rather than by a direct interaction of ER and PM in triads, as in the skeletal muscle. Therefore, in cardiomyocytes, the role of dystrophin in Ca^2+^ signalling may be different and indirect, or a non-structural role rather than a structural interaction cannot be excluded. Yet, the final outcome, i.e., the elevated Ca^2+^ concentration in the cytosol, is the same in both.

A reduced ability of Ca^2+^ removal from the cytosol to the SR/ER calcium stores caused by the significant reduction of SERCA activity is the most important anomaly in dystrophic cardiac cells. Inhibition of this calcium pump might be due to the increased amount of sarcolipin and a reduced proportion of phosphorylated phospholamban, which were detected in hearts of DMD patients and *mdx* mice [84]. The dephosphorylated form of phospholamban inhibits SERCA, thus a ratio between phosphorylated and dephosphorylated protein is an important factor modulating SERCA activity. The reduced amount of calcium stored in SR/ER in dystrophic cardiomyocytes may also be in line with the observed increased calcium leak through RyR2 [85]. This may contribute to an aberrant profile of calcium transients: decreased calcium peak in systole and reduced Ca^2+^ reuptake in diastole, resulting in the reduced heart contractility in systole. Taken together, excess of calcium originates from intracellular stores because of hyperactivity of RyR2. It has also been suggested that abnormal Ca^2+^ entry into cells may be due to the altered kinetics of LTCC/DHPR voltage-gated channels leading to their delayed inactivation and therefore improper interaction with mitochondria and finally affected metabolic response] [86]. Moreover, it occurs via increased stretch-activated channels (SACs) activity or sarcolemma micro-tears [54,56]

The significance of membrane tears as a cause of elevated Ca^2+^ entry into dystrophic cells is uncertain, given its applicability just to the mechanically challenged cells. Tears are unlikely to occur in dystrophic neurons, yet these cells also show elevated calcium (see below). On the other hand, abnormal shear stress response could perhaps explain elevation of Ca^2+^ in dystrophic endothelial cells. As mentioned earlier, abnormalities in these cells are puzzling also because of the unclear molecular mechanism. Endothelial cells are not known to express full-length dystrophin and therefore should not be affected by the *mdx* mutation. Yet, angiogenesis in these mice is seriously altered and impaired vascularisation affects regenerating muscles [87].

### Muscles That Get Away

Whatever the mechanism, the importance of altered calcium homeostasis for the dystrophic pathology is further illustrated by the fact that DMD oculomotor muscles are spared, which coincides with these muscles not having increased intracellular calcium levels [88]. If extraocular muscles are indeed spared because they have a mechanism managing their calcium dysregulation, it might be possible to mitigate the impact of the altered calcium homeostasis in the absence of dystrophin and specific treatments augmenting such a mechanism could have a great therapeutic value in DMD. A further credibility to this notion is added by the results of fluoxetine treatment in dystrophic *Sapje* zebrafish [39]. In this study, fluoxetine completely reversed dystrophic muscle damage in the total absence of dystrophin, just by improving calcium homeostasis via calsequestrin levels.

So far, most of the research has been focused on skeletal, cardiac but also on the brain consequences of DMD, reflecting their significance for the morbidity and mortality of this disease and also because these tissues express detectable levels of Dp427 protein.

## 8. Calcium Abnormalities in the Dystrophic CNS

Neuropsychological abnormalities in DMD are clinically important. While all patients are affected, one-third of boys manifest with severe cognitive and behavioural defects. This more severe phenotype has been associated with the loss of short dystrophins, which are the most abundant brain isoforms. Yet, it must be stressed that neuropsychological abnormalities have been identified in the absence of Dp427 only, in both patients and *mdx* mice. Furthermore, comparative proteomic profiling in *mdx-4cv* Dp427-deficient mouse brains revealed significant alterations in the Ca^2+^ binding protein calretinin and the Ca^2+^-pumping protein PMCA2, providing unbiased evidence that abnormalities of Ca^2+^ handling mechanisms occur in the absence of Dp427 [89]. These results confirm that *the loss of full-length dystrophins is both necessary and sufficient to trigger calcium defects in various cells of the CNS*, just as it is in skeletal and cardiac muscles.

Yet, in contrast to its uniform distribution under the sarcolemma of muscle fibres, 427 kDa dystrophin in the brain is specifically located at the postsynaptic regions of the cortical, hippocampal and cerebral Purkinje neurons, as well as in glial end-feet [34,35,90,91,92,93,94]. Although the biochemical basis of Ca^2+^ signalling in the brain is the same as in other tissues, the phenotypic effects of dystrophy are specific. It is unsurprising, given the diversity of cells which express dystrophin in the CNS. For example, intracellular Ca^2+^ mishandling was found in *mdx* cerebellar granule cells [95]. Moreover, elevation in [Ca^2+^]c correlating with an increased Ca^2+^ release from intracellular stores, such as ER, was observed in pyramidal cortical and hippocampal neurons [42]. Electrophysiological data [96] showed an increased post-tetanic potentiation, which might also be linked to calcium-regulatory defects [97]. Therefore, dystrophin is currently believed to modulate synaptic terminal integrity, distinct forms of synaptic plasticity and regional cellular signal integration, while Ca^2+^ is a second messenger in information integration in neurons [98,99,100,101]. However, in contrast to the progressive pathology in muscle cells, the increased neuronal Ca^2+^ levels detected as early as at day 24, did not lead to neuronal cell loss even in 9-month-old *mdx* mice [102]. Ca^2+^ dys-homeostasis also occurs in human DMD neurons [103], in line with alterations observed in skeletal and cardiac muscles [37,43,104]. Significant structural and molecular alterations were shown in neurons obtained by differentiation of hIPSc from a DMD patient affected by the cognitive impairment [103]. Moreover, DMD neurons showed a cytoplasmic Ca^2+^ overload associated with increased expression of the SERCA2 pump. Surprisingly, the decreased Ca^2+^ release peaks upon treatment of neurons with SERCA2 blocker (CPA) in Ca^2+^-free conditions, which indicates reduced levels of Ca^2+^ stored in ER. This result did not correlate with the increased expression of SERCA and could suggest that the intraluminal Ca^2+^ level in the dystrophic ER is lower than in the controls. This has, in fact, been reported in muscle [105].

Besides neurons, astrocytes generated from DMD hIPSc displayed defects in Ca^2+^ handling and a defective glutamate homeostasis [106].

Cognitive impairment in DMD has been associated with changes in the GABAergic system [99,107,108]. As GABA agonists can trigger a rise in [Ca^2+^]c, which could be inhibited by ryanodine, it suggests that Ca^2+^ release from intracellular stores might be involved [109]. The consequence of altered GABA signalling may be the facilitation of NMDA receptor-dependent synaptic plasticity [96]. Altered functionality of GABAergic neurons may also be responsible, or at least contribute, to the GABA-mediated trophic effects during neuronal development [108].

More recently, specific alterations in the expression of glutamatergic and P2 × 7 receptors were identified in the dystrophin-null mice cerebella [35], a finding consistent with the more severe CNS dysfunction occurring in patients with mutations affecting the short dystrophins.

Yet another causative factor in the cognitive impairment might be the dysregulation of the DAP complex, with associated aquaporin isoform 4 (AQP4) water and Kir 4.1 potassium channels, in perivascular glial end-feet. This is associated with a leaky blood–brain barrier (BBB) in dystrophic brains [110,111,112]. Given the importance of the intact BBB for the regulation of calcium concentrations between the extracellular and intracellular CNS compartments, the chronic Ca^2+^ overload in neurons and astrocytes, combined with impaired BBB buffering, may be intertwined with the defective synaptic transmission, resulting in the neurobehavioral impairment. Importantly, this dystrophic BBB abnormality is also caused by the absence of Dp427: in *mdx* glia, Dp71, albeit reduced, was still present [112]. However, further loss of short dystrophins exacerbated BBB permeability, as found in *mdx^βgeo^* mice [110,111].

## 9. Dystrophic Mitochondria and Altered Calcium Homeostasis

In all cell types, mitochondria are in the centre of calcium homeostasis. First, they produce cellular ATP, which is necessary to cover the energy demand of calcium transport against the concentration gradient [46,113]. Second, mitochondria transiently buffer an excess of cytosolic calcium, locally modifying its concentrations and thus influencing local calcium-dependent processes. Moreover, the elevation of Ca^2+^ concentration within physiological range stimulates mitochondrial energy metabolism and ADP phosphorylation. Thus, calcium homeostasis and mitochondrial energy metabolism are mutually dependent [114,115]. In resting cells, Ca^2+^ entry to the mitochondrial matrix is slow because of the low affinity of Ca^2+^ uniporter (MCU), while Ca^2+^ removal via mitochondrial Na^+^/Ca^2+^ exchanger (NCX) is relatively fast. This prevents mitochondrial Ca^2+^ accumulation. Therefore, in unstimulated cells, mitochondria do not play a role of calcium stores. However, upon excitation, when a local cytosolic Ca^2+^ concentration increases, mitochondria may transiently buffer an excess of this cation [113]. Conversely, an excessive Ca^2+^ uptake by mitochondria under pathological conditions of cellular calcium overload leads to an accelerated ROS generation, opening of the mitochondrial permeability transition pore (MPTP), activation of mitochondria-initiated apoptosis and collapse of the cellular energy metabolism [47,116,117].

### 9.1. Cardiac Mitochondria

Interestingly, such disturbances, which are fatal for most cells, in cardiomyocytes might lead to a progressive adaptation of mitochondria, which increase their Ca^2+^ uptake ability [118]. The elevated mitochondrial Ca^2+^ accumulation that was observed in dystrophic cardiomyocytes may be considered as a compensatory effect, partially replacing reduced calcium storage capability in the SR/ER system. It has been suggested that more efficient mitochondrial calcium uptake in dystrophic cardiomyocytes may occur because of increased levels of the channel forming subunit of the mitochondrial calcium uniporter (MCU) [118]. Moreover, levels of the mitochondrial Na^+^/Ca^2+^ exchanger are also increased, thus mitochondrial Ca^2+^ clearance is stimulated. On the other hand, very high cytosolic Ca^2+^ concentration counteracts calcium extrusion via NCX. Mitochondria in dystrophic cardiomyocytes were speculated to be more resistant to Ca^2+^-induced damage because of a reduced tendency for opening the mitochondrial permeability transition pore (MPTP) [118]. The MPTP opening due to excessive mitochondrial calcium levels enhances mitochondrial ROS production and depolarisation of the mitochondrial membrane, which are considered important in the cardiac pathology [119]. Interestingly, DMD-associated changes in cardiac mitochondria are noticeable before the clinical symptoms of this disease [55,118].

### 9.2. Skeletal Muscle Mitochondria

In dystrophic skeletal muscle, mitochondrial MPTP seems to be more sensitive to Ca^2+^ and therefore prone to be opened with increased Ca^2+^ concentrations. While the amounts of protein components of the pore might be altered, these changes were not consistent [118]. Moreover, the Ca^2+^ buffering capacity of dystrophic mitochondria was lower than in wild type muscles. Interestingly, a reduced MCU activity in isolated *mdx* skeletal muscle mitochondria has been found [118], most likely due to the altered subunit composition of the MCU complex. Importantly, this stands in clear contrast to the aforementioned MCU increases in cardiomyocyte mitochondria [120]. Furthermore, in contrast to the cardiac data, mitochondria isolated from skeletal muscles did not show any changes in the NCX content.

### 9.3. Structural Alterations

Apart from the alterations in the amount and activity of mitochondrial calcium-handling proteins, aberrant intracellular architecture of mitochondrial networks and mitochondria–ER interactions have a huge impact on cellular calcium handling and its abnormalities in dystrophic cells. Mitochondria and the ER membranes form sub-compartments known as mitochondrial associated membranes (MAMs), which contain specific calcium toolkit proteins and are crucial for the regulation of Ca^2+^ release from ER and for its mitochondrial uptake. These contact sites are directly engaged in the modulation of calcium signals and in the activation of mitochondrial energy metabolism. It was found that the density of MAMs in dystrophic cardiomyocytes is increased, while in skeletal muscles it is lower than in their wild type equivalents.

Aberrant organisation of the mitochondrial and ER networks may result from altered protein profiles at the mitochondria–ER junction and from modified cytoskeletal organisation due to lost interactions between dystrophin and actin filaments. It is plausible that pharmacological modulation of such intra-organelle contacts may be a target for DMD therapy. However, changes in the mitochondrial network architecture were also observed in *mdx* myoblasts [74] as well as in cells derived from DMD patients [121,122], which complicates the picture.

## 10. Impact of Therapies on Calcium Handling

Normalisation of calcium handling in dystrophic muscles has long been tested as a potential therapeutic approach which could ameliorate DMD progression, and a number of drugs were tested. As many have been summarised in an excellent review [123] these examples will not be repeated here. However, several less-known pharmaceutical treatments illustrating the key points discussed in previous chapters will be considered.

The calcium channel blockers showed little impact on the DMD pathology [64], indicating that entry via these channels may only play a secondary role in this abnormality. However, there is evidence that other channel blockers can ameliorate the phenotype.

Among drugs which directly influence Ca^2+^ homeostasis those causing inhibition of Ca^2+^ channels in the sarcoplasmic membrane, stabilization of ryanodine receptors in SR/ER, overexpression of SERCA and reduction of the opening probability of MPTP were found to be of significance [124,125]. In a recent study, treatment of *mdx* mice with SERCA activator reduced cytosolic Ca^2+^ levels, restored mitochondrial function, enhanced muscular strength, reduced muscular degeneration and fibrosis. Importantly, this treatment prevented exercise-induced muscular damage in the absence of dystrophin (doi:10.1093/hmg/ddab100).

Metformin treatment normalised MAMs and modulated mitochondrial metabolism in *mdx* cardiac cells [126]. Although a beneficial effect of metformin on dystrophic cells needs further confirmation, its positive impact on a spectrum of metabolic processes is well established [127,128].

Deflazacort, which belongs to the class of glucocorticoids used in DMD therapy as anti-inflammatory, is likely to influence SOCE and affect other calcium toolkit proteins. It was shown to enhance mitochondrial efficiency in *mdx* skeletal muscle cells [129]. However, molecular mechanisms behind these effects and their possible impact on DMD progression are still elusive.

An interesting example of an approach aimed at the destabilisation and rupture of the dystrophic sarcolemma is the copolymer-based treatment. Application of these membrane-interacting synthetic molecules, known also as poloxamers or pluronics, to *mdx* cardiomyocytes in vitro ameliorated the calcium overload. Moreover, in *mdx* mice in vivo, P188 improved cardiac functions [130]. However, the same copolymer not only failed to prevent sarcolemma leakage in *mdx* skeletal muscle but even had some deleterious effects on skeletal muscle functions [131,132]. These cardio-specific effects of pluronics may be explained by very significant differences in the physico-chemical properties of the sarcolemma in cardiac and skeletal muscles or that their mechanism of action is different than as a membrane sealant.

While there are still many questions regarding the pathomechanism(s) of DMD cognitive impairment, improvements resulting from dystrophin re-expression in *mdx* brains [133] indicate that, even in adults, some correction could be achieved. Unfortunately, none of the currently tested approaches addresses this significant burden for DMD patients and their families.

In this respect, a pathological commonality between muscle and cognitive dysfunctions might be expedient. P2RX7 is one of the main drivers of both DMD muscle damage and inflammation [134], as well as the key mediator of inflammatory processes within the nervous system [135]. Therefore, P2RX7 appears to be such a common link. Indeed, ablation and pharmacological inhibition of this purinoceptor alleviated muscle symptoms [136,137] and also improved recognition memory and diminished anxiogenic-like behaviours [138]. Importantly, the key effect of the activation of this ATP-gated ion channel permeable to divalent cations is the increase of intracellular calcium levels [139,140]. The P2RX7 blockade appears to be the first clinically applicable comprehensive treatment for DMD. It is also yet another example that therapies exploiting more accessible targets downstream from the absence of dystrophin offer a better chance of success [141]. In fact, the entire extracellular ATP-mediated signalling network might be such a target [28].

Gentamycin, later supplanted by the more specific Ataluren, initiate a read through premature termination codons and therefore synthesis of full-length dystrophin. In the dystrophic skeletal muscles, gentamycin improved the direct interaction between voltage-gated channels and RyR, which occurs upon cell excitation. In *mdx* smooth muscle, in which calcium signalling abnormalities are less severe than in skeletal muscle fibres, gentamycin treatment stimulated RyR expression and restored CICR-based activation of calcium signalling [142]. Exon skipping restores a reading frame and expression of truncated but functionally active mini-dystrophin [143]. Interestingly, exon-skipping produced very similar effects to those evoked by gentamycin [142]. Thus, therapeutic approaches leading to re-expression of the full length or mini-dystrophin both led to improved calcium homeostasis in muscle cells in the same experimental paradigm.

## 11. Are There Calcium Abnormalities in BMD?

As explained in the Introduction, BMD is a milder variant of muscular dystrophy caused by mutations that preserve the reading frame and result in truncated but semi-functional mini- or micro-dystrophins being expressed. The BDM progress, albeit slower than in the case of DMD, may also lead to disability and premature death. Surprisingly, we failed to identify in the literature any studies that directly addressed the effects of BMD mutations on the Ca^2+^ homeostasis. Some indirect data indicate that changes in calcium signalling in BMD cells should be less severe. For example, only limited impact of BMD on mitochondrial metabolism suggests that mitochondrial Ca^2+^ buffering capacity as well as ATP delivery are not affected [144]. Furthermore, reports that re-expression of mini-dystrophin, either through gene therapy or exon skipping, resulting in the conversion of DMD into BMD, have been shown to stabilise calcium homeostasis. Specifically: exon 45 skipping normalised Ca^2+^ signals in cardiomyocytes derived from IPSc [145]; Dystrophin re-expression via exon-skipping prevented the excessive Ca^2+^ load and myotube damage in golden retriever dogs [146] and in *mdx* mice [142]. In yet another experiment, transfection of dystrophic muscle fibres with a mini-dystrophin-encoding gene prevented an abnormal Ca^2+^ efflux from SR through the inositol trisphosphate receptor (IP3R) [52]. Expression of mini-dystrophin in *mdx* muscle restored the abnormal Ca^2+^ entry through channels [51]. Thus, expression of mini-dystrophins appears to ameliorate the dystrophic calcium overload, indicating that binding sites in the rod domain may be largely dispensable for maintaining calcium homeostasis.

## 12. Concluding Remarks

Although often attempted, DMD pathology cannot be currently explained by a single mechanism. It is likely that specific, often overlapping, sometimes distinct processes are at play in different tissues and at different stages of the disease. However, abnormalities of calcium homeostasis are the common pathological feature found in both DMD patients and all animal models of this disease and in all dystrophic cells, irrespective whether electrically excitable or non-excitable. Mutations affecting the full-length dystrophin expression are both necessary and sufficient for the calcium phenotype to occur, while the loss of short dystrophins may exacerbate it. However, the role of dystrophin in calcium signalling is not completely understood, which may explain why no treatment targeting calcium pathways has been successful so far.

In dystrophic cells, the net outcome is always the elevated resting cytosolic Ca^2+^ concentration, but the biochemical mechanisms leading to that outcome may have components specific to a particular cell type. The intensity of calcium responses in dystrophic cells results from the aberrant, stimulus-dependent interplay between proteins involved in maintaining calcium homeostasis, and these differ across cells.

In skeletal muscles, cardiomyocytes, and neurons, the DAP complex, properly assembled only in the presence of dystrophin, appears to serve as a scaffold for proteins engaged in calcium homeostasis, while dystrophin interactions with the actin cytoskeleton influence ER organisation and motility. However, in some cells, calcium abnormalities cannot be linked to the increased sensitivity to mechanical stimuli or even clearly attributed to the loss of interaction between dystrophin and the calcium toolbox proteins. For example, myoblasts, lymphocytes, and endotheliocytes, despite expressing dystrophin transcripts, are not known to synthesise detectable levels of Dp427 protein. Yet, *DMD* gene mutations lead, in these cells, to significant calcium abnormalities. In fact, already in the developing dystrophic stem cells (including ESC), in mesenchymal and myogenic cells, significant calcium anomalies are not only present but appear to contribute to the early pathology.

Given that the impairment of calcium homeostasis seems to underpin multiple DMD abnormalities in a plethora of dystrophic cells, understanding mechanisms behind this alteration may lead to the development of better therapeutic approaches to treat this devastating and incurable disease.

## Figures and Tables

**Figure 1 ijms-22-11040-f001:**
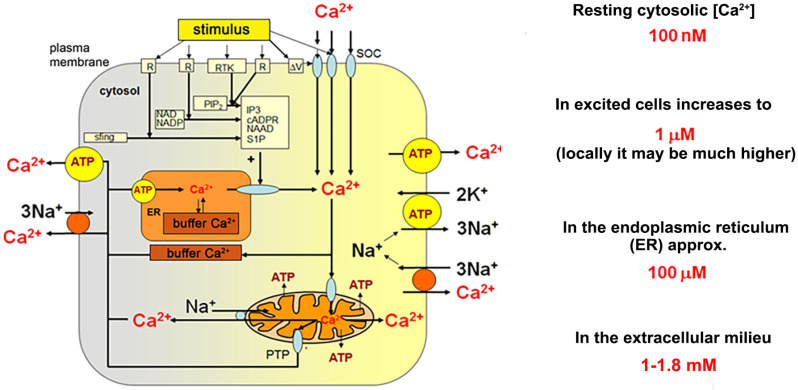
Key elements of cellular calcium homeostasis in an animal cell. (According to Berridge et al. (2000) [40] Modified).

**Figure 2 ijms-22-11040-f002:**
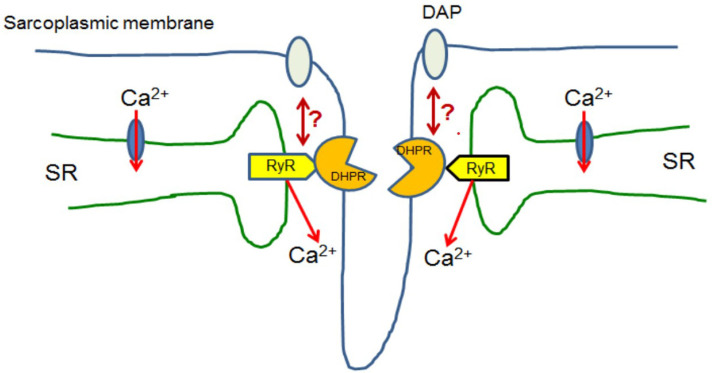
Skeletal muscle. Upon excitation, skeletal myofibres L-type calcium channel (DHPR/LTCC) proteins localised in selected regions of the sarcoplasmic membrane interact with ryanodine receptors (RyR) localised in the specific regions of the sarcoplasmic reticulum (SR). Such structures are known as triads. Ca^2+^ released by these RyRs activates muscle contraction and then is transported back to SR. Presumably, dystrophin associated complex indirectly interacts with this mechanism, but its role is still unclear [45].

## Data Availability

Not applicable.

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
