# Peer review of "Disrupted Calcium Homeostasis in Duchenne Muscular Dystrophy: A Common Mechanism behind Diverse Consequences"

_ijms, 2021, doi:10.3390/ijms222011040_

Round 1

Reviewer 1 Report

Role of calcium mishandling in Duchene Muscle dystrophy (DMD) has been frequently reported and the underlying mechanism is discussed im Detail in many studies. Hence, the submitted manuscript is a follow-up of already known reports  on Calcium homeostatis and Duchene muscle dystrophy. Having Said that, the draft includes ample  Information at one Place. The manuscript  is nicely written with analysis and synthesis of published  works addressing the role and subsequent mechanism of Calcium homeostatis disruption in Duchene muscle dystrophy. After thoroughly going  through the manuscript, I would suggest the authors to address following Points:

  1. A Brief introduction on DMD genetics and how DMD is different from Becker Muscle dystrophy (BMD) which is associated with the mutations on the same gene.
  2. Is there any mechanistic difference in Calcium homeostatis disruption in DMD and BMD?
  3. The thereupatic Strategies that are known so far in Warding off/Mending Calcium homeostatis disruption and their effectiveness need to be briefly discussed.

Author Response

We would like to thank you for your insightful and inspiring comments. Below you can find our responses. We hope these  are satisfying and meet the expectations.

Reviewer: A Brief introduction on DMD genetics and how DMD is different from Becker muscular dystrophy (BMD) which is associated with the mutations on the same gene. Authors’ response: We have expanded the introduction with specific sections on the DMD genetics and the genotype-phenotype correlation and also added a specific section on BMD genetics and its pathological impact.

Reviewer:  Is there any mechanistic difference in Calcium homeostasis disruption in DMD and BMD?

Authors’ response: We are very grateful for this suggestion, as following thus suggestion,  the additional information enriched the review significantly. In the improved manuscript we discuss differences between DMD and the allelic variant Becker muscular dystrophy, with a particular focus on a few studies on calcium abnormalities between these two diseases. We also explain the impact of therapeutic re-expression of Becker-type mini-dystrophins on calcium abnormalities.

Reviewer:  The therapeutic strategies that are known so far in Warding off/Mending Calcium homeostasis disruption and their effectiveness need to be briefly discussed.

Authors’ response: As suggested, we discuss the data on the effectiveness of various  therapeutic interventions in improving the calcium homeostasis. For this, we added a dedicated section to the manuscript. The “Mechanisms of calcium homeostasis” chapter and relevant figures have been revised.

Reviewer 2 Report

In this manuscript, authors provided lots of information of DMD, mostly focus on Dp427. I have several comments as follow,

  1. In introduction, there are many disorganized paragraphs. Please reorganize these paragraphs to introduce the common knowledge of DMD, the genotype-phenotype correlation, and the focus part of this review.
  2. In the section “Mechanisms of calcium homeostasis”, after the figure and figure legend, there are two paragraph A and B, what is the correlations of these two paragraphs and their subtitles? Are they figure legends or text? Calcium signalling toolkit can be observed in all kinds of cells or only in skeletal muscle?
  3. I suggest that Figure 1A and 1B can be separated to be two figures, because they were described in two separate sections.
  4. On page 6, authors mentioned “The functional or structural link between dystrophin and specific proteins more or less directly involved in calcium signalling has been suggested quite early, but a precise mechanism of such interactions has remained unknown [44-47,27,48,35]. The earliest theory explaining calcium overload in dystrophic muscle fibres had implicated destabilisation and rupture of the sarcolemma in the wake of muscle contraction. Although still re-called, this concept is currently questioned and displaced by other hypotheses (see below).”. Please more specific describe what is the earliest theory and other hypotheses. This paragraph needs more information involved in.
  5. On page 7, authors mentioned “Given that these abnormalities were found in the established mdx myoblast cell line, they are cell-autonomous and not triggered by the dystrophic niche.” Please describe what kinds of abnormalities were observed in cell line.
  6. Does this review (from section 2-7) all focus on Dp427?
  7. Similar to introduction, there are many paragraph in each section, please organize them and mention the key information you want to show.

Author Response

In this manuscript, authors provided lots of information of DMD, mostly focus on Dp427. I have several comments as follow:

Reviewer: In introduction, there are many disorganized paragraphs. Please reorganize these paragraphs to introduce the common knowledge of DMD, the genotype-phenotype correlation, and the focus part of this review.

Authors’ response: It has been significantly restructured,  as suggested

Reviewer: In the section “Mechanisms of calcium homeostasis”, after the figure and figure legend, there are two paragraph A and B, what is the correlations of these two paragraphs and their subtitles? Are they figure legends or text? Calcium signalling toolkit can be observed in all kinds of cells or only in skeletal muscle?

Authors’ response: We agree with the Reviewer. This section has been revised significantly and all these points has been clarified in the new version of the manuscript

Reviewer: I suggest that Figure 1A and 1B can be separated to be two figures, because they were described in two separate sections.

Author s’ response: It has been corrected in the revised version of the manuscript.

Reviewer: On page 6, authors mentioned “The functional or structural link between dystrophin and specific proteins more or less directly involved in calcium signalling has been suggested quite early, but a precise mechanism of such interactions has remained unknown [44-47,27,48,35]. The earliest theory explaining calcium overload in dystrophic muscle fibres had implicated destabilisation and rupture of the sarcolemma in the wake of muscle contraction. Although still re-called, this concept is currently questioned and displaced by other hypotheses (see below).”. Please more specific describe what is the earliest theory and other hypotheses. This paragraph needs more information involved in.

Authors’ response: In the revised version of the manuscript this paragraph has been completely restructured and expanded to clarify the point that we are trying to make . On page 4 of the revised version of the manuscript this issue is explained as follows:

“Importantly, the mechanism(s) triggering muscle death have not been fully elucidated, although “the sarcolemmal hypothesis” put forward in 1990s’ has become the leading one and it weighs heavily on the therapeutic approaches. It stems from the findings on the dystrophin-DAP interactome abnormalities and can explain many clinical findings. This dominant view on the pathogenesis od DMD states that in the absence of dystrophin anchoring the DAP, this complex is not properly attached in the myofiber sarcolemma and that its loss destabilizes the cell membrane, which becomes  torn during contractions. Moreover, impaired architecture of intracellular scaffolds affects the appropriate localisation and function of various other proteins important for myofiber operations [23].

Yet, several studies, rather unexpectedly, challenged this widely accepted view of dystrophin anchoring DAP in the myofibre sarcolemma being a key to muscle health. It has been shown that ablation of dystrophin in fully differentiated myofibres did not trigger DAP dysregulation or muscle degeneration [24], while ablation of both dystrophin and DAP only triggered a mild decrease in muscle contractile force but not muscle degeneration [25]. Astoundingly, there is even evidence that myofibres can function entirely without dystrophin, just with overexpression of specific genetic modifiers [26]. Furthermore, cell-autonomous defects are noticeable in DMD cells, which, when healthy, express the 14 kb DMD transcript but do not produce detectable levels of full-length dystrophins [27-30]. These, undoubtedly surprising findings, shed a new light on DMD pathophysiology (see below)”.

Reviewer: On page 7, authors mentioned “Given that these abnormalities were found in the   

established mdx myoblast cell line, they are cell-autonomous and not triggered by the dystrophic niche.” Please describe what kinds of abnormalities were observed in cell line.

Authors’ response: Your again, we agree that this point was not explained clearly. Now, itis described on page 10, as follows:

“Until recently, potential DMD abnormalities in myoblasts were not investigated. The reason for this was that healthy myoblasts do not express detectable dystrophin protein. Therefore, no abnormalities have been foreseen. Against this reasonable prediction, in proliferating myoblasts, mutations in the DMD gene cause multiple changes in calcium signalling [30,28]. Next generation sequencing (NGS) and subsequent biochemical analyses revealed changes in the expression of a number of genes encoding calcium signalling and related proteins (e.g., STIM 1, SERCA1 and 2A, calsequestrin, PLC variants 3 and 4, NCX 1 and 2, PMCA, Gaq11, IP3R [28]. Moreover, numerous functional abnormalities including elevated SOCE activity [30], altered responses of both metabotropic and ionotropic receptors [29,28], and reduced oxygen consumption, together with stimulated glycolysis [74] distinguish immortalized mdx myoblasts from their wild type (w/t) equivalents”.

Reviewer: Does this review (from section 2-7) all focus on Dp427?

Authors’ response: Yes. The reason or this approach has been explained in more detail on page 3, as follows:

“The DMD gene mutation hotspots are located in the regions affecting the full-length isoforms. Indeed, mutations leading to the loss of full-length 427 kDa dystrophin are both necessary and sufficient for causing the muscle pathology. The CNS abnormalities seem to be further exacerbated by the loss of the short isoforms, Dp71 and Dp140, which are highly expressed in the normal brain. There is little data on the muscle impact of mutations affecting these short isoforms or on the dystrophin-null patients. In this review we have therefore focused primarily on the canonical DMD caused by the absence of Dp427”

Reviewer: Similar to introduction, there are many paragraph in each section, please organize them and mention the key information you want to show.

Authors’ response: As suggested by the Reviewer, the paragraphs have  been modified and distinct sections were given titles.

The authors are very grateful to the Reviewer for their insightful and helpful comments and questions. We believe that the Reviewers’ input allowed as to focus and improve the manuscript significantly.

Round 2

Reviewer 2 Report

No extra comments.